# SDG26 Is Involved in Trichome Control in *Arabidopsis thaliana*: Affecting Phytohormones and Adjusting Accumulation of H3K27me3 on Genes Related to Trichome Growth and Development

**DOI:** 10.3390/plants12081651

**Published:** 2023-04-14

**Authors:** Jing Zeng, Lanpeng Yang, Minyu Tian, Xiang Xie, Chunlin Liu, Ying Ruan

**Affiliations:** 1Key Laboratory of Hunan Provincial on Crop Epigenetic Regulation and Development, Hunan Agricultural University, Changsha 410128, China; 2Key Laboratory of Crop Physiology and Molecular Biology of Ministry of Education, Hunan Agricultural University, Changsha 410128, China; 3School of Energy and Environment and State Key Laboratory of Marine Pollution, City University of Hong Kong, Kowloon, Hong Kong 999077, China

**Keywords:** *Arabidopsis thaliana*, histone methylation, SDG26, trichomes, H3K27me3, plant hormone, gene expression

## Abstract

Plant trichomes formed by specialized epidermal cells play a role in protecting plants from biotic and abiotic stresses and can also influence the economic and ornamental value of plant products. Therefore, further studies on the molecular mechanisms of plant trichome growth and development are important for understanding trichome formation and agricultural production. SET Domain Group 26 (SDG26) is a histone lysine methyltransferase. Currently, the molecular mechanism by which SDG26 regulates the growth and development of Arabidopsis leaf trichomes is still unclear. We found that the mutant of Arabidopsis (*sdg26*) possessed more trichomes on its rosette leaves compared to the wild type (Col-0), and the trichome density per unit area of *sdg26* is significantly higher than that of Col-0. The content of cytokinins and jasmonic acid was higher in *sdg26* than in Col-0, while the content of salicylic acid was lower in *sdg26* than in Col-0, which is conducive to trichome growth. By measuring the expression levels of trichome-related genes, we found that the expression of genes that positively regulate trichome growth and development were up-regulated, while the negatively regulated genes were down-regulated in *sdg26*. Through chromatin immunoprecipitation sequencing (ChIP-seq) analysis, we found that SDG26 can directly regulate the expression of genes related to trichome growth and development such as *ZFP1*, *ZFP5*, *ZFP6*, *GL3*, *MYB23*, *MYC1*, *TT8*, *GL1*, *GIS2*, *IPT1*, *IPT3*, and *IPT5* by increasing the accumulation of H3K27me3 on these genes, which further affects the growth and development of trichomes. This study reveals the mechanism by which SDG26 affects the growth and development of trichomes through histone methylation. The current study provides a theoretical basis for studying the molecular mechanism of histone methylation in regulating leaf trichome growth and development and perhaps guiding the development of new crop varieties.

## 1. Introduction

The trichome structure and its derivatives evolved with the evolution of plants. Trichomes can be found on the leaves, stems, and flowers of some gymnosperms, bryophytes, and most angiosperms. Trichomes play an important role in plant growth and defense against biotic and abiotic stress [1,2]. Trichome is a first line of defense, protecting plants before pathogens or herbivores attack [3,4,5,6]. In addition, trichomes also play an important role in protecting plants by inducing pollination, resisting extreme environments, protecting against ultraviolet radiation, heavy metals, drought, salinity, and preventing biological invasion and mechanical damage [7]. Therefore, plants with trichomes are more advantageous in herbivore-rich environments and arid regions [3,8]. Trichomes arise from the proliferation of epidermal cells that undergo cell division, differentiation, and growth to produce the epidermal extension of the tissue. In cells, the morphology of trichomes is diverse, including glandular, non-glandular, branched, or non-branched [9,10]. Usually, unicellular trichomes have a simple structure without glands. Most angiosperms, some gymnosperms, and bryophytes are non-glandular trichomes [11]. Trichomes also have a significant impact on the quality and yield of commercial crops. Cottonseed wool is an important raw material for the textile industry [12,13]. The trichomes of tea leaves are rich in nutrients and are crucial for tea breeding and tea quality [14]. The light-leaf and light-shell phenotypes of crop rice have missing trichomes, which are beneficial for crop harvesting and subsequent processing [15]. Therefore, an in-depth study of the molecular mechanisms of trichome growth and development is necessary. The trichomes of Arabidopsis are unicellular, non-glandular hairs, which are excellent objects for studying the molecular mechanisms of plant trichome development. Existing research has established a model of the growth and development mechanism of Arabidopsis trichomes [16,17]. Arabidopsis trichomes mostly occur on rosette leaves, stems, cauline leaves, and sepals, but not on hypocotyls and cotyledons [18]. Arabidopsis trichomes develop from a single pro-epidermal cell at the leaf base [19]. Its growth and development are coordinated by many factors, such as the environment, hormones, regulatory genes, and non-coding factors [20,21,22]. Studies have shown that in Arabidopsis, many transcription factors are involved in the initiation, growth, and development of plant trichomes. The positive regulatory factors include the Transparent Testa Glabra1 (TTG1) proteins of the WD40 family. Four basic Helix-Loop-Helix transcription factor (bHLH)-like transcription factors GLABRA3 (GL3), Enhancer of Glabra3(EGL3), Transparent Testa (TT8) and MYC1. Three R2R3MYB-related transcription factors, Glabra1 (GL1), MYB Domain Protein 23 (MYB23) and MYB Domain Protein 5 (MYB5), are involved in trichome development. [23,24,25,26,27,28,29]. They then form the MYB-bHLH-TTG complex and bind to the *GLABRA2* (*GL2*) gene promoter [30]. The negative regulators include Caprice (CPC), Tripty Chon (TRY), Enhancer of TRY and CPC1 (ETC1) and Enhancer of TRY and CPC2 (ETC2). The genes of these transcription factors are functionally redundant and negatively regulate trichome development through repression of *GL3*, *EGL,3* and *TTG1* [31]. Studies have shown that hormones play a key role in the growth and development of trichomes, including gibberellin (GA), cytokinin (CK), salicylic acid (SA), jasmonic acid (JA), and brassinosteroids Lactone (BR) [32,33,34,35]. Among them, GA, CK, JA, and brassinolide promote the occurrence of trichomes, while SA inhibits the growth and development of trichomes by interfering with the jasmonate pathway [36,37,38,39,40].

Histone modification, a chromatin modification that regulates gene expression, has been recognized as a crucial epigenetic mechanism in many organisms [41]. It is widely involved in multiple biological processes, such as gene expression, growth and development, and the stress response of plants. Histone modifications are post-translational covalent modifications to the amino-terminal tail of nucleosomes, including methylation, acetylation, phosphorylation, ubiquitination, and sumoylation [42,43]. Histone H3 lysine methylation in Arabidopsis usually occurs at four sites: lysine 4 of histone 3 (H3K4), lysine 9 of histone 3 (H3K9), lysine 27 of histone 3 (H3K27), and lysine 36 of histone 3 (H3K36). There are differences in the number of methyl groups added to the same amino acid at the same site, such as monomethylated (me1), dimethylated (me2), and trimethylated (me3) [42,44]. Among them, H3K4me3 and H3K36me3 are generally considered to be markers of activating genes, and H3K9me2 and H3K27me3 are generally considered to be repressive markers [45,46,47,48]. Histone methylation is mediated by histone methyltransferases. Histone lysine methyltransferases all have a common domain—the SET domain. SDG26 is a histone methyltransferase. SDG26 mainly regulates the accumulation of H3K4me3, H3K36me3, and H3K27me3 on the chromatin of regulatory genes [49]. The current research on SDG26 mainly focuses on the regulation of plant flowering, and some studies have also studied the relationship between SDG26 and UV-B damage and DNA repair [50,51,52]. Recent studies have found that SDG26 is one of the members of the complex autonomous Pathway Complex (AuPC). In addition to SDG26, it also contains FLD (Flowering Locus D), LD (Luminidependens), EFL2, EFL4, and APRF1. Mutants of the AuPC complex possessed increased histone modifications associated with transcriptional activation (H3Ac, H3K4me3, and H3K36me3) and decreased histone modifications associated with transcriptional repression (H3K27me3) [53].

Our previous study found that the *sdg26* mutant possessed more trichomes than Col-0 in the natural growth process, speculating that SDG26 significantly affected the growth and development of Arabidopsis trichomes [44]. However, the regulatory mechanism of SDG26 on the Arabidopsis trichomes is still unclear. In order to further understand the effect of histone methylation on the Arabidopsis trichomes. We observed and statistically analyzed the phenotypes of *sdg26* and Col-0 at different growth stages and measured their hormone content and the expression levels of genes related to trichomes. Chromatin immunoprecipitation-sequencing analysis was performed on both *sdg26* and Col-0 to elucidate the molecular mechanism by which histone methyltransferase SDG26 regulates the growth and development of Arabidopsis trichomes. The main purpose is to elucidate the molecular mechanism by which histone methyltransferase SDG26 regulates the growth and development of Arabidopsis trichomes.

## 2. Results

### 2.1. Effects of SDG26 on Arabidopsis Trichomes

We observed the trichome phenotypes of Arabidopsis at different vegetative growth stages (5-, 9-, and 13-leaf stages) and found that at the middle and late stages of vegetative growth, *sdg26* individuals were larger than Col-0 (Figure 1A). The area and number of trichomes on the leaves of *sdg26* were also greater than those of Col-0 leaves (Figure 1B). At the 5-leaf stage, the number of total trichomes and trichome density of *sdg26* were 44.3% and 42.3% more than those of Col-0, respectively (*p* < 0.001). At the 9-leaf stage, the leaf area and trichome density of *sdg26* were 47%, 52.2%, and 20.4% larger than those of Col-0, respectively (*p* < 0.001). At the 13-leaf stage, *sdg26* had 59.6%, 66.8%, and 21.2% more leaf area, the total number of trichomes, and trichome density than Col-0, respectively (*p* < 0.001) (Figure 1C–E).

We further counted the number of trichomes per unit area (1 cm^2^) and the ratio of different morphological trichomes of *sdg26* and Col-0 at different growth stages (Figure 2A). The results showed that the proportions of the trichomes in the two branches of Col-0 were 44.5%, 81.6%, and 81.2% more than those of *sdg26* at the 5-, 9-, and 13-leaf stages, respectively (*p* < 0.001). The proportions of trichomes in the three branches of *sdg26* were 8.7% (*p* < 0.01), 15.4%, and 14.3% (*p* < 0.001) more than those of Col-0, respectively. The proportion of trichomes in the four branches of *sdg26* was not significantly different from that of Col-0 (Figure 2B–D). With the growth and development of Arabidopsis leaves, the number of trichomes per unit area decreased, but the number of trichomes in *sdg26* was still significantly more than Col-0 in each period. The number of trichomes per unit area of *sdg26* was 32.9%, 41.5%, and 34.9% more than Col-0 at the 5-, 9-, and 13-leaf stages, respectively (*p* < 0.001) (Figure 2E). In summary, dysfunctional SDG26 promotes the occurrence of Arabidopsis trichomes and has a positive effect on stabilizing the morphological development of trichomes.

### 2.2. Effects of SDG26 on Plant Hormones

Many plant hormones directly affect the occurrence of plant leaf trichomes. Hormone signals are mainly involved in the formation of trichomes by mediating the expression of downstream genes. Among them, CK and JA can positively promote the growth of trichomes, while SA can inhibit the growth of trichomes [15,54].

In order to determine whether SDG26 affects plant hormones that regulate the growth of trichomes, we measured the contents of four CK in different growth stages of the mutants *sdg26* and Col-0. The four CK are riboside isopentenyladenosine (iPR), N6- (2-isopentenyl) adenine (iP), trans-zeatin riboside (tZR), and trans-zeatin (tZ). At the same time, the contents of JA and SA were also determined. At the 5-leaf stage, iPR, iP, tZR, tZ, and JA in *sdg26* were 40.9%, 47.1%, 54.2%, 63.5%, and 25.7% higher than those in Col-0. The SA content in Col-0 was 46.6% higher than that in *sdg26*. At the 9-leaf stage, iPR, iP, tZR, tZ, and JA in *sdg26* were 38.1%, 56.4%, 40.5%, 40.6%, and 25.3% higher than those in Col-0. SA in Col-0 was 42.3% higher than that in *sdg26*. At the 13-leaf stage, iPR, iP, tZ, and JA in *sdg26* were 50.4%, 41.6%, 22.2%, and 47.8% higher than those in Col-0. The tZR and SA acid contents in Col-0 were 26.9% and 15.3% higher than those in *sdg26,* respectively (Figure 3). The results showed that the contents of CK and JA, which promote the growth and development of trichomes, were significantly increased in *sdg26*, while the contents of SA, a hormone that inhibits the growth and development of trichomes, were significantly decreased in *sdg26*, thus promoting trichome growth and development on the leaves of *sdg26*. Taken together, a loss-of-function of SDG26 leads to increased levels of CK and JA and decreased levels of SA, thereby promoting the development of trichomes in Arabidopsis leaves.

### 2.3. Expression of Genes Affecting Trichome Growth and Development in SDG26

To further explore the molecular mechanism by which SDG26 regulates trichomeogenesis, we wished to identify its putative target genes (Figure 4). We measured the expression of genes related to trichome growth and development (*CPR5*, *EGL3*, *TTG1*, *GIS*, *GL1*, *GL3*, *SAD*, *MYB23*, *MYC1*, *TT8*, *CPC*, *ETC1*, *ETC2*, *ETC3*, *TCL1*, and *TRY*) and genes related to plant hormone regulation of trichome development (*ZFP1*, *ZFP5*, *ZFP6*, *ZFP8*, *IPT1*, *IPT3*, *IPT5*, *CKX3*, *CKX4*, *CKX5*, *COI1*, and *JAZ1*) were detected. As shown in Figure 4, at the 5-, 9-, and 13-leaf stages, the expression of genes *CPR5*, *EGL3*, *TTG1*, *GL1*, *GL3*, *SAD*, *MYB23*, *MYC1*, *ZFP1*, *ZFP5*, *ZFP6*, *IPT3*, and *IPT5* was significantly up-regulated (*p* < 0.05). In addition, some genes were up-regulated at some stages, such as *IPT1*, which was significantly up-regulated at the 5- and 13-leaf stages (*p* < 0.05), *GIS* and *TT8*, which were significantly up-regulated at the 9- and 13-leaf stages (*p* < 0.05), and *COI1*, which was only significantly up-regulated at the 13-leaf stage up-regulated (*p* < 0.05). The expression levels of genes *CPC*, *ETC1*, *ETC2*, *JAZ1*, *TCL1*, *TRY*, *CKX4*, *CKX3*, and *CKX5* that had a negative effect on trichome development were significantly down-regulated (*p* < 0.05). *ETC3* was significantly down-regulated at the 9- and 13-leaf stages (*p* < 0.05). In summary, in sdg26, the coordination between the up regulation of positive regulatory genes and the downregulation of negative regulatory genes is more conducive to the formation of trichomes.

### 2.4. Levels of H3K27me3 at the Chromatin Sites of Trichome-Related Genes

To determine which genes are directly regulated by sdg26 modification, we performed ChIP-seq. Statistics on the distribution of histone modification-enriched regions on chromosomes revealed that a total of 39,908 peaks were detected in sdg26 and 44,900 peaks were detected on Col-0. 3163 differential peaks were found between sdg26 and Col-0 (Appendix A). Through functional enrichment analysis of genes associated with differential histone modifications, it was found that the signal transduction pathways related to plant hormones were significantly enriched (Appendix A). This suggests that the effect of SDG26 on plant hormones may be related to signal transduction. We tracked the levels of H3K27me3 in selected genes. The results showed that the enrichment of H3K27me3 on the chromatin of genes *ZFP1*, *ZFP5*, *ZFP6*, *GL3*, *MYB23*, *MYC1*, *TT8*, *GL*, *GIS*, *IPT1*, *IPT3*, and *IPT5*, which are positive regulators of trichome development, decreased in *sdg26* (Figure 5). H3K27me3 accumulation is mainly associated with repression of gene expression [48]. In *sdg26*, the accumulation of H3K27me3 on genes related to the promotion of trichome development was reduced, leading to the upregulation of the expression of these genes and thereby promoting the development of Arabidopsis trichomes.

## 3. Discussion

SDG26 is a member of the Arabidopsis trxG histone lysine methyltransferase family. Previous research has shown that SDG26 regulates plant flowering time [44,50]. The recent study found that SDG26 can form a complex with other proteins to reduce the accumulation of the H3K27me3 histone modification associated with transcriptional repression [53,55]. Currently, there is little reported research on the effects of SDG26 on leaf trichomes. Here, we first found that the density of trichomes per unit area of *sdg26* leaves was significantly higher than that of Col-0 at different growth stages (Figure 1). Second, the proportion of epidermal trichomes with two branches in *sdg26* was significantly lower than that of Col-0, while the proportion of epidermal trichomes with three branches was significantly higher than that of Col-0. Third, there was no significant difference in the proportion of epidermal trichomes with four branches between *sdg26* and Col-0 (Figure 2). These results indicate that SDG26 regulates the trichome density and the ratio of bifurcated trichomes to trifurcated trichomes.

According to previous research, we have learned that not only are many transcription factors involved in regulating the initiation, growth, and development of trichomes, but plant hormones also have a significant impact on them. To further explore the molecular mechanisms by which SDG26 regulates trichomes, we measured the CK, JA, and SA levels of *sdg26* and Col-0 at different growth stages. It was found that the content of CK and JA, which can promote trichome development, was significantly higher in *sdg26* than that of Col-0, while the content of SA, which inhibits trichome development, was significantly lower in *sdg26* than Col-0. It indicated that the loss of SDG26 function increased the levels of hormones (CK and JA) that promoted the growth and development of trichomes and decreased the levels of hormones (SA) that inhibited the growth and development of trichomes (Figure 3). We further measured the expression of genes related to the growth and development of trichomes and genes related to plant hormone regulation of trichome. We found that the genes that positively regulate growth and development of trichomes were significantly upregulated in *sdg26*, with the expression levels of *COI1*, *CPR5*, *EGL3*, *TTG1*, *GL1*, *GL3*, *GIS*, *SAD*, *MYB23*, *MYC1*, *TT8*, *ZFP1*, *ZFP5*, *ZFP6*, *IPT1*, *IPT3*, and *IPT5* all significantly upregulated (*p* < 0.05). The expression levels of genes with negative effects on trichome growth and development, such as *CPC*, *ETC1*, *ETC2*, *ETC3*, *JAZ1*, *TCL1*, *TRY*, *CKX3*, *CKX4*, and *CKX5*, were significantly downregulated (*p* < 0.05) (Figure 4). According to these results, we found that SDG26 affects the growth and development of trichomes in two main ways. First, it affects the synthesis and degradation of cytokinins, thereby affecting the content of cytokinins and indirectly affecting the growth of trichomes. *IPT1*, *IPT3* and *IPT5* are key genes for cytokinin synthesis, while *CKX3*, *CKX4* and *CKX5* are key genes for cytokinin degradation [56]. Second, SDG26 affects the regulatory network during trichome development.

According to the analysis of ChIP-seq results in this study, we found that the accumulation of H3K27me3 in the chromatin regions of *ZFP1*, *ZFP5*, *ZFP6*, *GL3*, *MYB23*, *MYC1*, *TT8*, *GL*, and *GIS* among the *sdg26* upregulated genes was lower than that of Col-0 (Figure 5). These results indicated that the loss of function of SDG26 led to a decrease in H3K27me3 accumulation on chromatin regions associated with some C2H2 family members, R2R3-MYB, and bHLH transcription factors and promoted the upregulation of their genes expression, thereby enhancing the initiation of trichomes. Simultaneously, our observations revealed a diminished accumulation of H3K27me3 within the chromatin domains of *IPT1*, *IPT3*, and *IPT5* in the *sdg26* compared to the Col-0 (Figure 5). The IPT family of proteins represents crucial rate-limiting enzymes in the biosynthesis of cytokinins in plants. The *atipt1357* mutant exhibits substantial reductions in the concentrations of isopentenyladenine, trans-zeatin (tZ), and their corresponding nucleosides, nucleotides, and glucosides, resulting in inhibited growth [57]. Consequently, the altered cytokinin content within *sdg26* is attributed to the decreased accumulation of H3K27me3 in the chromatin domains of *IPT1*, *IPT3*, and *IPT5* (Figure 3). This evidence suggests that SDG26 may modulate trichome growth and morphogenesis by influencing cytokinin pathways. SDG26 tends to form a histone modification complex with FLD and LD to regulate the expression of the FLC gene, thereby further affecting the flowering time of plants [55]. In the latest research reports, it was also verified that SDG26 possesses no histone methyltransferase activity when it is a member of the AuPC complex. Mutants lacking any of the components of the AuPC complex showed decreased histone modifications associated with transcriptional repression (H3K27me3) [53]. Therefore, in our study, SDG26 may also adjust H3K27me3 accumulation on the chromatin of target genes through the complex and then realize the regulation of the growth and development of Arabidopsis trichomes.

Based on the experimental results, a model was established for how SDG26 regulates trichome development (Figure 6). SDG26 regulates the content of CK by affecting key rate-limiting enzymes such as *IPT1*, *IPT3*, and *IPT5* in the CK synthesis pathway. At the same time, SDG26 regulates the expression of C2H2 transcription factors *ZFP1*, *ZFP5*, *ZFP6*, and *GIS*, thereby regulating the reception and transduction of CK signals by C2H2 transcription factors. The key trimeric complex activator MBW in the developmental process of trichome initiation is partially regulated by SDG26. The expression of *TT8*, which can promote MBW, is also regulated by SDG26. SDG26 regulates the growth and development of trichomes by regulating the expression of *MYC1* and affecting the expression of *TTG2* and *GL2*. Taken together, our study reveals the mechanism by which SDG26 regulates trichome development in Arabidopsis by adjusting H3K27me3 accumulation level on relative genes, which provides new research ideas and strategies for genetic improvement of crop quality.

## 4. Conclusions

This study focuses on the regulation of growth and development of trichomes by histone lysine methyltransferase SDG26 in Arabidopsis. We found that there are two main aspects that cause the difference in trichome density between *sdg26* and Col-0 by analyzing the phenotypes, hormone levels, gene expression levels, and ChIP-seq. On the one hand, it is related to the content of plant hormones. In *sdg26*, gene expression is conducive to cytokinin synthesis, and cytokinin is conducive to trichome growth. On the other hand, it is related to the regulatory network during trichome development. In *sdg26*, the expression of genes that positively regulate the growth and development of trichomes is up-regulated, while the expression of negatively regulated genes is down-regulated. We found that in *sdg26*, the accumulation of H3K27me3 on the chromatin of genes regulating cytokinin synthesis and positively regulating the growth and development of trichomes is lower than that in Col-0. In summary, histone lysine methyltransferase SDG26 participates in the regulation of growth and development of trichomes by regulating the accumulation of H3K27me3 on genes in Arabidopsis.

## 5. Materials and Methods

### 5.1. Plant Material and Growth Condition

All Arabidopsis thaliana seeds from mutant and wild-type strains used in this work are from the Columbia (Col-0) ecotype background. Seeds of *sdg26* (SALK_013895) [43,58]. Seeds of Col-0 and *sdg26* were surface sterilized and allowed to grow on sterilized nutrient soil. Subsequently, the plates were cold treated at 4 °C for 3 days without light exposure and then kept in a climate-controlled growth chamber at 22 °C with 65% humidity under a light intensity of 100 μmol m^−2^ s^−1^ and long-day conditions (16 h of light and 8 h of darkness) [59].

### 5.2. Morphologic Analysis

A camera was used to record the second true leaf of Arabidopsis at the 5-leaf stage, the 9-leaf stage, and the 13-leaf stage. The leaf color, saturation, brightness, and other parameters were adjusted through the RGB color function of the ImageJ software to make the color of the leaves uniform. The color threshold was adjusted to select the darkest area of the leaves, and the Limit to Threshold function in the Analyze-Set Measurements menu was utilized to calculate the area of the selected image [60,61]. The second true leaves of Col-0 and *sdg26*, which were growing at different growth stages, were photographed with a stereomicroscope, and the total number of trichomes on the leaves were counted. A 1 cm^2^ area on the leaf was chosen, and the number and branching situation of the trichomes were documented. The number of samples for each determination was 30 strains, and three sets of biological replicates were carried out.

### 5.3. Quantification of Hormone

Arabidopsis leaf samples were ground using liquid nitrogen. Then, 100 mg of the sample was weighted into a pre-cooled centrifuge tube, and 1.5 mL of the extraction solution (50% acetonitrile with 0.3 ng of 4′-Hydroxyazobenzene-2-carboxylic acid as an internal standard) was added; this was then sonicated for about 3 min in an ice bath, and then placed in a rotating instrument overnight at 4 °C. The next day, the sample was centrifuged at 13,000 rpm for 10 min at 4 °C, and the supernatant was transferred into a new 2 mL centrifuge tube. 1 mL of the extract was added to the precipitate and extracted again at 4 °C for 60 min, centrifuged, and the supernatant was combined with the last supernatant [62].

The content of CK in the leaves of *Arabidopsis* at different stages was determined by an internal standard method. The phytohormone standard was dissolved with methanol to obtain a 1 mg/mL phytohormone standard stock solution for future use. The stock solution was serially diluted to concentrations of 100 μg/mL, 10 μg/mL, and 1 μg/mL (the working solution). The working solution was prepared as a standard solution for establishing a standard curve, and its concentrations were 0.5 ng/mL, 5 ng/mL, 10 ng/mL, 30 ng/mL, 60 ng/mL, and 100 ng/mL.

According to the properties of plant hormones, Waters BEH C18 RP Column (1.7 μm, 2.1 × 150 mm) liquid chromatography was used. The chromatographic system used was the ExionLC AD System from SCIEX. Oasis HLB RP (1 mL/30 mg) cartridges were washed sequentially with 1 mL of methanol and 1 mL of water, then equilibrated with 50% ACN.The effluent was collected by adding the extract to an Oasis HLB RP column, washed with 30% (*v*/*v*) ACN, and combined with the previous effluent. The effluent was blown dry with nitrogen gas at low temperature, then resuspended in 300 μL of 30% acetonitrile and filtered through a 0.22 μm nylon membrane, and the obtained filtrate was used for mass spectrometry analysis. Automatic identification and integration were performed with MultiQuant (AB Sciex, Massachusetts, USA) software. A linear regression standard curve was drawn with the peak area of the mass spectrum of the analyte as the ordinate and the mass of the analyte as the abscissa. The sample concentration calculation was performed through substitution of the mass spectrum peak area of the sample analyte into the linear equation to calculate the mass of the hormone in the sample. This study measured four types of cytokinins, jasmonic acid, and salicylic acid.

### 5.4. Quantitative RT-PCR

For details about the total RNA extraction, refer to TAKARA RNAiso Plus^TM^ instruction manual. The cDNA was synthesized using the kit provided by Fermentas. The synthesized cDNA was diluted at a concentration of 1:10 and stored for later use. The required primers were downloaded from the qPrimerDB-qPCR Primer Database (accessed on 19 May 2021. https://biodb.swu.edu.cn/qprimerdb) or quantitative PCR primers were designed in a targeted manner according to the basic principles of quantitative PCR primer design (Supplemental Appendix A). The experimental operation was performed according to the operation method provided by the SYBR^®^ Select Master Mix Real-time Fluorescence Quantitative Kit. The expression level of each gene is displayed relative to the reference gene ACT2, and the relative expression level is analyzed according to the relative Ct value method according to the finally obtained Ct value, and statistical analyses were conducted with one-way ANOVA [63].

### 5.5. ChIP-Seq

Formaldehyde and PMSF were added to the Fix Buffer (0.4 M sucrose, 10 mM Tris-HCl, and 1 mM EDTA). 100 mL of the Fix Buffer was used to fix the sample, and 20 strains of Col-0 and *sdg26* were placed into the Fix Buffer and vacuumed for 15 min to deflate and restore the pressure. The vacuuming step was repeated until the sample turned dark green, indicating that the Fix Buffer had penetrated. 5 mL of 2.5 M glycine was added to 100 mL of Fix buffer, mixed well, and vacuum pumped for 10 min to stop the fixation. The sample was washed 3–5 times with ultrapure water, blotted dry with paper, and weighed [44]. The samples were put into Falcon tubes, flash-frozen in liquid nitrogen, and stored at −80 °C. ChIP-seq was performed by E-GENE Tech Co., Ltd. (Shenzhen, China). Extracted nuclei were resuspended in lysis buffer (0.1% SDS, 10M EDTA, and 50 mM Tris-HCl, pH 8.0) and sonicated through a Bioruptor Pico to generate chromatin fragments ranging in size from 100–800 bp. An aliquot of the chromatin sample was sonicated as an input control for isolation. The remaining chromatin was diluted with the ChIP dilution buffer (1% Triton, 2 mM EDTA, 150 mM NaCl, and 20 mM Tris-HCl, pH 7.5), and antibodies were incubated with 4 μg protein and A/G magnetic beads (Invitrogen; 10003D). Immunoprecipitation reactions were incubated overnight at 4 °C and then washed at 4 °C with a low-salt wash buffer (20 mM Tris-HCl, pH 8.0, 150 mM NaCl, 2 mM EDTA, 0.1% SDS, and 1% Triton X-100). Beads were washed twice for 5 min each using a high salt wash buffer (20 mM Tris-HCl, pH 8.0, 500 mM NaCl, 2 mM EDTA, 0.1% SDS, and 1% Triton X-100) and a LiCl wash buffer (10 mM Tris-HCl, pH 8.0, 0.25 M LiCl, 1 mM EDTA, 1% NP-40, and 1% sodium deoxycholate) and were soaked at 4 °C for 5 min. Finally, reactions were washed twice with TE buffer. Captured DNA and input samples were reverse-crosslinked with elution buffer (1% SDS, 0.1 M NaHCO_3_) at 65 °C for 2 h. The eluted DNA was purified by a phenol-chloroform extraction, and the resulting DNA was subjected to library preparation and sequenced on an Illumina sequencing platform. The ChIP assay was performed with an antibody against histone H3K27me3. MACS2 bdgdiff was used to identify differential peaks. Then the R package ChIPseeker was used to annotate the differential peaks. Performing functional enrichment analysis on the genes associated with the differential peaks helps to understand which biological processes or pathways the genes associated with the differential histone modification/transcription factor sites are mainly involved in. The raw ChIP-seq data have been submitted to NCBI SRA under the project numbers: PRJNA948834 and PRJNA949206.

## Figures and Tables

**Figure 1 plants-12-01651-f001:**
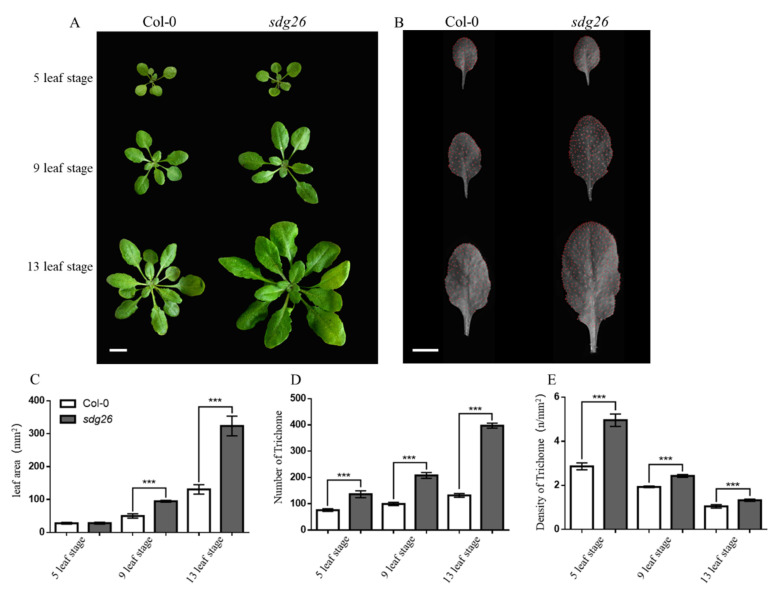
Effect of SDG26 on Arabidopsis Trichomes (**A**): The overall phenotype of rosette leaves of Arabidopsis at different stages. Scale bar = 3 cm. (**B**): Phenotype of the second true leaf and distribution of trichomes in Arabidopsis at different stages. Scale bar = 1 cm. (**C**): Statistics of the second true leaf area of Arabidopsis. (**D**): The total number of trichomes in the second true leaf of Arabidopsis. (**E**): Density of trichomes in the second true leaf of Arabidopsis. Error bars represent the SD (n = 3); *** represents *p* < 0.001.

**Figure 2 plants-12-01651-f002:**
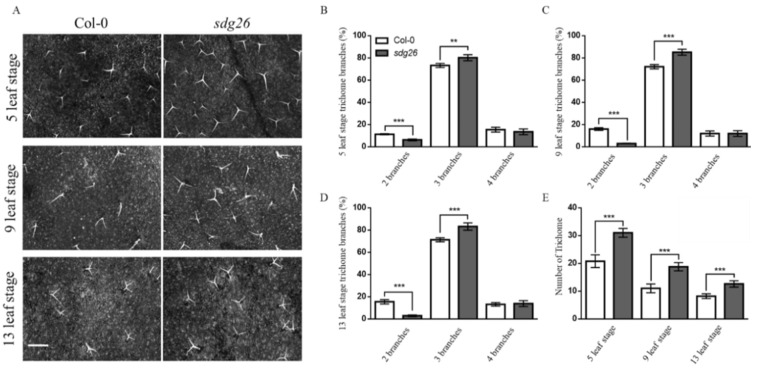
The number of epidermis cells per unit area of Col-0 and *sdg26* and the classification statistics of different morphological trichomes in different periods. (**A**): The growth of the trichomes of the second true leaf of Col-0 and *sdg26*. Scale bar = 1 mm. (**B**–**D**): Proportions of different morphological trichomes on the second true leaves of Col-0 and *sdg26* at different growth stages, with a statistical area of 1 cm^2^. (**E**): The number of trichomes on the second true leaf of Col-0 and *sdg26* at different growth stages; the statistical area is 1 cm^2^. Error bars represent the SD (n = 3); ** represents *p* < 0.01, *** represents *p* < 0.001.

**Figure 3 plants-12-01651-f003:**
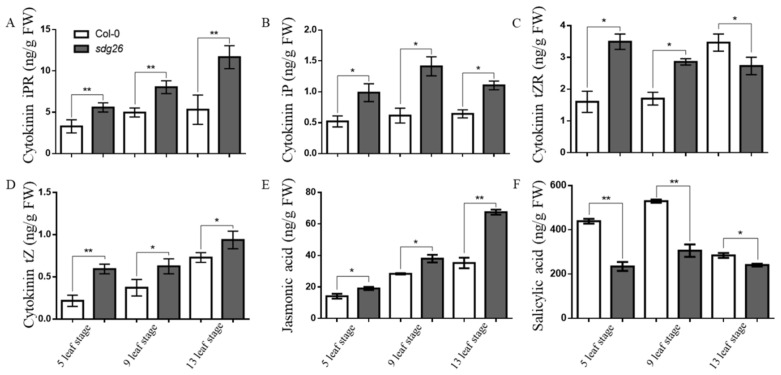
Hormone content in Col-0 and *sdg26* at different periods. (**A**–**F**): iPR, iP, tZR, tZ, JA, and SA contents of Col-0 and *sdg26* at different growth stages. Error bars represent the SD (n = 3); * represents *p* < 0.05, ** represents *p* < 0.01.

**Figure 4 plants-12-01651-f004:**
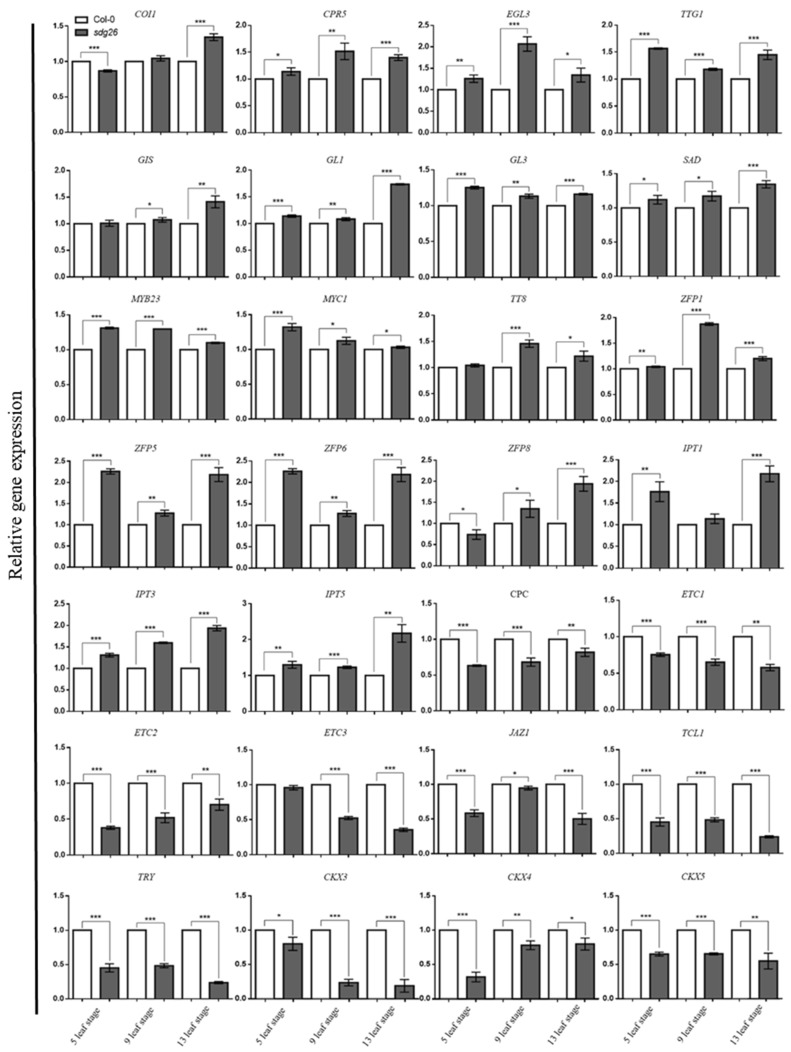
Expression of genes related to growth and development of trichomes in Col-0 and *sdg26* at different growth stages. Error bars represent the SD (n = 3); * represents *p* < 0.05, ** represents *p* < 0.01, *** represents *p* < 0.001.

**Figure 5 plants-12-01651-f005:**
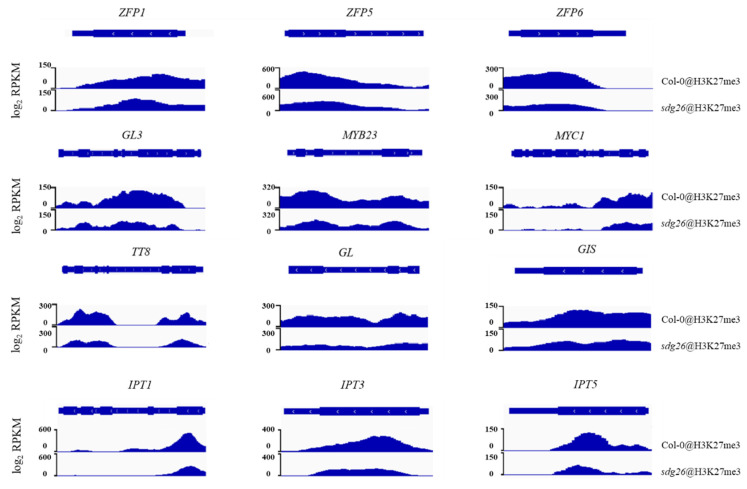
Integrative Genomics Viewer images showing the distribution of H3K27me3 enrichment levels in Col-0 and the *sdg26* mutant at the *ZFP1*, *ZFP5*, *ZFP6*, *GL3*, *MYB23*, *MYC1*, *TT8*, *GL*, *GIS*, *IPT1*, *IPT3*, and *IPT5* loci. RPKM: reads per kilobase per million mapped reads.

**Figure 6 plants-12-01651-f006:**
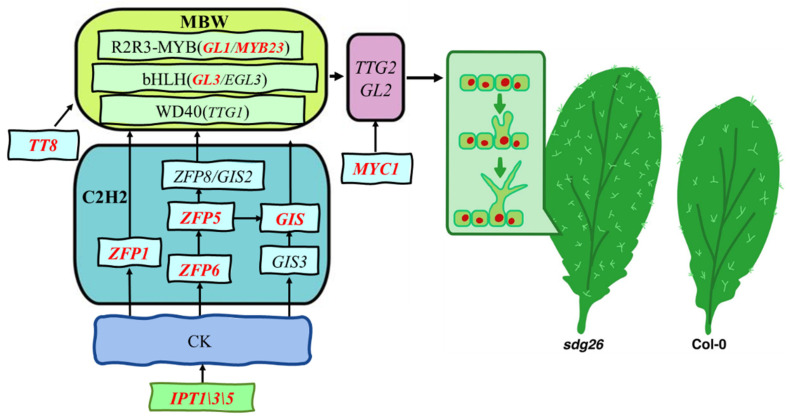
Pathway diagram of histone methyltransferase SDG26 regulating the growth and development of Arabidopsis trichomes. Black arrows indicate stimulatory effects, and red fonts indicate genes whose H3K27me3 accumulation is directly affected by SDG26.

## Data Availability

The raw ChIP-seq data have been submitted to NCBI SRA under the project numbers: PRJNA948834 and PRJNA949206.

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
