# Peer review of "SDG26 Is Involved in Trichome Control in Arabidopsis thaliana: Affecting Phytohormones and Adjusting Accumulation of H3K27me3 on Genes Related to Trichome Growth and Development"

_plants, 2023, doi:10.3390/plants12081651_

Round 1

Reviewer 1 Report

The authors described the overproduction of trichomes in the sdg26 mutant and studied the potential basis underpinning this phenotype. Based on the results presented by the authors, the higher number of trichomes in sdg26 was ascribed to the misregulation of hormone homeostasis and altered deposition of H3K27me3, which illustrated the role of histone methylation in trichome development. In this regard, the submitted work is of considerable novelty and significance.

 However, there are some minor issues in this manuscript. For example, the  “byin” in the subtitle of bullet 2.3 should be corrected. A scale bar should be included in Figure1. The ChIP-seq data should be deposited in a public database and the accession number should be stated in the revised manuscript. As the authors proposed that SDG26 regulated hormone homeostasis and H3K27me3 to orchestrate the development of trichomes, it is reasonable that some genes involved in hormone metabolism are epigenetically regulated, which the authors should further discuss in the revised manuscript.

 Collectively, I believe the submitted work can be of interest to general readership and has particular significance. Thus, I suggest this manuscript be published after minor revision.

Reviewer 2 Report

In this manuscript, the role of SDG26 in trichome formation and development was dissected at the metabolic, transcriptional, and epigenetic levels. The manuscript is well structured, and I only have the following minor comments/suggestions.

Title: « Thaliana » should be « thaliana »

Figure 1: Scale bars should be added

Introduction: “Three R2R3MYB-related transcription factors GLABRA1 (GL1), MYB DOMAIN PROTEIN 23 (MYB23) and MYB DOMAIN PROTEIN 5 (MYB5)”. A verb is missing in this sentence.

Results 2.1: “(Fig. 1C, D, F)” should be “(Figure 1C, D, E)”, I think.

2.3. Expression of genes affecting trichome growth and development byin SDG26. Please revise as needed.

Discussion: “Recent studies have shown that SDG26 tends to form a histone modification complex with FLD, SDG26 and LD to regulate”. To regulate what?

5.2 “Phylogenetic analysis” is not appropriate.

HABA abbreviation should be explained.
